# Physico-Chemical Study of Mn(II), Co(II), Cu(II), Cr(III), and Pd(II) Complexes with Schiff-Base and Aminopyrimidyl Derivatives and Anti-Cancer, Antioxidant, Antimicrobial Applications

**DOI:** 10.3390/molecules28062555

**Published:** 2023-03-11

**Authors:** Maged S. Al-Fakeh, Maha A. Alsikhan, Jawza Sh Alnawmasi

**Affiliations:** 1Department of Chemistry, College of Science, Qassim University, Buraidah 51452, Saudi Arabia; 2Taiz University, Taiz 3086, Yemen

**Keywords:** mixed ligands, Schiff-Base, anti-cancer, antioxidant, antimicrobial

## Abstract

A new class of biologically active mineral complexes was synthesized by reacting the following metal salts: MnCl_2_·4H_2_O, CoCl_2_·6H_2_O, CuCl_2_·2H_2_O, CrCl_3_·6H_2_O, and PdCl_2_ respectively with 2-amino-4,6-dimethyl pyrimidine (ADMPY) and Schiff’s base resulting from the condensation reaction between benzaldehyde with *p*-phenylenediamine and 2-hydroxy-1-naphthaldehyde as ligands have been synthesized and characterized on the basis of their CHN, thermal analysis, XRD, SEM and magnetic measurements along with their FT-IR and UV-vis spectra. The scanning electron microscope SEM measurements and the calculations on the powder XRD data indicate the nano-sized nature of the prepared complexes (average size 32–88 nm). The spectral data confirmed the coordinated ligand (HL) via a nitrogen atom of an azomethine group (-C=N-) and phenolic -OH group and NH_2_-ADMPY ligand with the metal ions. An octahedral geometry for all complexes has been proposed based on magnetic and electronic spectral data except Pd(II) complex, which has a tetrahedral geometry. Molecular modeling was performed for Cu(II) complex using the density functional method DFT/B3LYP to study the structures and the frontier molecular orbitals (HOMO and LUMO). The antioxidant of the complexes was studied using the 2,2-diphenyl-1-picrylhydrazyl (DPPH)-free radical-scavenging assays. The metal complexes were tested in vitro for anticancer activities against two cancer lines A-549 and MRC-5 cells. Cu(II) and Pd(II) complexes showed the highest cytotoxicity effect, comparable to that of other cis-platinum-based drugs. The complexes showed significant activity against fungi and bacteria.

## 1. Introduction

Recently, the focus has been on mixed chelated mineral complexes due to their importance and biological activity against microbes, fungi, cancerous tumors, and others. Schiff bases are the most common coordination compounds. They were named Schiff bases in relation to their discovery by the German scientist Hugo Schiff, who discovered them in 1864. They were simply synthesized by condensing primary amines and carbonyl compounds [1]. The azomethine (anils, imines) group (-HC=N-) is a structural feature shared by these compounds. This active group carries a potential binding site for metal ions through the lone pair of electrons (unlinked) in a hybridized orbital SP^2^ of the nitrogen atom. The (C=N) linkage is required for biological activity in azomethine derivatives; numerous azomethines have been reported to have strong antifungal, antibacterial, anticancer, and antimalarial properties. Numerous Schiff bases and their complexes have been investigated for their noteworthy and intriguing qualities, such as their photochromic qualities and capacity to bind to some toxic metals [2]. Here we find Schiff’s base acting as a toothed flexible ligand and coordination is carried out through the nitrogen atom of the azomethine group and the oxygen atom of the phenolic group. It is close to the azomethine group [3], so it was shown in studies that Schiff bases are excellent chelating agents, and they also provide a great attraction factor due to the ease of preparation, structural flexibility, and the special properties of the C=N group [4,5]. In addition, pyridine derivatives are among the important categories in inorganic chemistry, where 2-amino-4,6-dimethyl pyrimidine is one of its important derivatives, which plays an important vital role, as it participates in coordination. Pyrimidine is a member of the heterocyclic compound family and is a common substance found in living things. It holds a special position in the field of heterocyclic and medicinal chemistry because of its beneficial biological properties and therapeutic uses. Since the structural subunits of the pyrimidine compound are found in many naturally occurring products, including vitamins and antibiotics, it is widely distributed in nature and is extremely important to human health. Consequently, they pay close attention to the development of both biologically energetic compounds and antibiotics. Tri-substituted pyrimidine containing an electron-withdrawing group such as the amino group in pyrimidine shows more potent in vitro antimicrobial activity [6,7,8,9]. Thus, the synthesis and screening of Schiff base ligand, 2-((E)-((4-(((E)-benzylidene)amino)phenyl)imino)methyl)-naphthalene-1-ol with transition metal complexes ability, anticancer activity, antioxidant properties, and antibacterial properties would be of substantial biological importance [10]. Previously, complexes of Cr(III), Mn(II), Ni(II), Fe(III), and Zn(II) were reported to show anticancer, antioxidant, and antibacterial activity [10]. Therefore, we speculate that complexes of Mn(II), Co(II), Cu(II), Cr(III), and Pd(II) with 2-amino-4,6-dimethyl pyrimidine (ADMPY) and Schiff’s base will show more anticancer, antibacterial, and antioxidant activity. These findings prompted us to start a project on metal-ADMPY and Schiff’s base complexes, involving an investigation of their nano-sized, anticancer, antibacterial, and antioxidant properties. The main objective of this research is to prepare and study the physico-chemical composition of the new nano-sized complexes of manganese(II), cobalt(II), copper(II), chromium(III), and palladium(II), which are studied in solution and in the solid state. The stability constants are evaluated and structure of the studied complexes is elucidated using elemental analyses, FT-IR, UV-vis spectra, magnetic moment, molar conductance, and thermal analyses measurements. In the anticancer studies, antibacterial, antifungal, and antioxidant properties of the metal complexes are reported. The Schiff base ligand used in this study are HL (Figure 1a) and ADMPY (Figure 1b). 

## 2. Results and Discussion

The reactions of manganese(II), cobalt(II), copper(II), chromium(III), and palladium(II) with HL and (ADMPY) ligands proceed readily to form the corresponding mixed ligand complexes. The metal mixed-bonding complexes were respectively prepared at a ratio of M:L1:L2 (1:1:1). The complexes are air-stable, insoluble in common organic solvents but soluble in dimethylformamide (DMF) and dimethylsulphoxide (DMSO). The molar conductivities of the complexes in DMSO at 25 °C are in the 11.4–28.2 Ω^−1^cm^2^ mol^−1^ range indicating the non-electrolytic nature of the compounds [11,12,13]. The following figure (Figure 2) illustrates these suggested summary formulas.

### 2.1. Fourier Transform Infrared Spectra

The IR spectra provide important information about the nature of the functional group attached to the metal. The IR spectrum of the free ligands is compared to the spectra of the complexes to study the bonding mode of Schiff base and (ADMPY) ligands to metal complexes. Table 1 shows the main IR bands and their assignments. FTIR spectrum of ligand (HL) did not show the stretching frequency of phenolic (O-H) because of its involvement in hydrogen bonding (intra-ligand hydrogen bond) with lone electron pair of an azomethine nitrogen atom (O-H……N) [14]. The υ(O-H) vibrations of complexes **1**, **2**, **3**, **4** and **5** were observed as strong bands at 3410, 3345, 3438, 3326, and 3354 cm^−1^ respectively. Strong bands appear at 1618, 1604 cm^−1^ belonging to the two azomethine groups υ(C=N) [15]; it also shows a medium band of the υ(C=C) at 1540 cm^−1^ [16]; the stretching frequency of the phenolic υ(C-O) bond has been recorded as 1310 cm^−1^. The spectrum of ligand (ADMPY) observed the medium intensity absorption bands 3396 cm^−1^ and 3310 cm^−1^ are due to asymmetric and symmetric vibrations of υ(NH_2_) stretching frequency, respectively, while the bending band-specific scissors at 1633 cm^−1^ appears [17], C-N. It is found in two places; the base ring and the primary amine group. Comparisons were made by contrasting the compound’s spectrum with published literature on similar systems. It was found that the stretching frequency of the phenolic υ(C–O) bond has been shifted in spectra of prepared complexes of the υ(C-O) stretching band, which provides evidence that the phenolic-OH group is involved in chelation. This chelation is further reinforced also by the appearance of new peaks in complexes due to δ(OH) rocking [18], its blue-shift indicates the sharing of phenolic oxygen atom in coordination with metal ions after losing its proton. The azomethine group υ(C=N) in all complexes undergoes a shift, necessitating its coordination with the metal center. Because the NH_2_ group of the ADMPY ligand will be overlapping with those of the water molecules, the presence of water coordination in the complexes’ spectra makes it challenging to conclude them. Clarifying how chelation affects the in-plane bending, or δ(NH_2_) vibration, adds to the evidence that the NH_2_ group is involved. This band’s shift from 1633 cm^−1^ in the free ADMPY ligand to 1667–1698 cm^−1^ in the complexes shows that the NH_2_ group was involved in complex formation [19]. The υ(OH) in the lattice H_2_O in complexes (**2**) and (**4**) may be ascribed to a broad diffused band with medium intensity situated at 3547 and 3548 cm^−1^ respectively [20,21]. The coordinated H_2_O υ(OH) stretching vibration for the complexes (**1**) through (**5**) appears at a wavelength of 3302–3330 cm^−1^ ranges in each case. A band at 732–747 cm^−1^ in the IR spectra of the complexes **1**–**5** appears as (H_2_O), this suggests that coordinated water is present [22]. New bands are found in the spectra of the complexes in the regions 640–612 cm^−1^ which are assigned to υ(M-O). The bands at 566–544 cm^−1^ have been assigned to υ (M-N) of the amino mode [23]. The υ(M-Cl) bands appeared at 442–419 cm^−1^ [24].

### 2.2. Magnetic Moments

At room temperature, the magnetic moment (*μ*eff) values for the complexes (**1**–**4**) are given in Table 2, and the results are 5.84 B.M, 4.90 B.M, 1.72 B.M, and 3.70 B.M. respectively. Mn(II) and Co(II) complexes are 5.84 and 4.90 B.M, this indicates high-spin. Mn(II) and Co(II) ions have five and three unpaired electrons in each compound’s respective outer valence shell [25,26]. The magnetic moment of the Cu(II) complex was 1.72 BM, which is consistent with an octahedral shape and one unpaired electron [27]. Indicating the presence of three unpaired electrons in its outer valence shell and creating an octahedral geometry around the Cr(III) ion, the Cr(III) complex displayed a magnetic moment of 3.70 BM [28]. The diamagnetic behavior of the Pd(II) complex suggests a square-planar geometry [29,30].

### 2.3. Electronic Spectra

Electronic spectra of metal complexes and free ligands (HL), (ADMPY) use DMSO (10^−3^ M) as solvent (Table 2). The ligand UV–Vis spectrum (HL) peaks at 259 nm are assigned to (π→π∗) [10]. The ligand (ADMPY) showed absorbance bands at 314 assigned to (*n*→π∗) [6]. As for the electronic spectra of complexes for Mn(II), Co(II), Cu(II), Cr (III), and Pd(II), it has distinct bands in the visible region of spectra attributed to d-d transitions ranges at 415–455 nm. As for the bands corresponding to the π→π∗ and *n*→π∗, transitions appeared in the ranges of 252–308 nm and 300–355 nm, respectively, which indicates a bonding between the ligand and the metal elements. We may suggest the following chemical formulae for the produced five metal mixed ligand complexes based on the previously collected data elemental analyses, infrared and electronic spectra, magnetic moment measurements, and molar conductance measurements (Figure 3).

### 2.4. Theoretical Study 

We carried out the DFT optimization through PBE1PBE functional using 6–31G (d)/Lanl2dz basis sets. The copper center is five coordinated (Figure 4) and bonded to the first ligand with Cu–Ni (2.02 Å) and Cu-OH (2.23 Å). It is also bonded to 2 Cl (2.32 Å and 2.32 Å). For the second ligand, it is bonded to NH_2_ with a distance Cu-NH_2_ = 2.13 Å. The electron density is distributed on the first ligand for the HOMO (Figure 5) but it is mainly distributed on the metal for the LUMO.

### 2.5. Thermal Analysis

The DTA and TGA curves in the temperature range 20–550 °C for the complexes appear at a heating rate of 10 °C/min; they showed an agreement in weight loss between their results obtained from the thermal decomposition and the calculated values, which supports the results of elemental analysis and confirms the suggested formulae. TG analysis results of mixed ligand complexes are summarized in Table 3, it was possible to determine the following characteristic thermal parameters for each reaction step: Initial point temperature of decomposition (Ti): the point at which TG (Tm) curve starts deviating from its base line. Final point temperature of decomposition (Tf): the point at which TG curve returns to its base line.

#### [Pd(HL)(ADMPY)(H_2_O)]·Cl_2_

One water molecule’s loss (calculated at 2.69%, found at 2.51%) and the observed mass loss of this initial stage are closely correlated. Two chlorine atoms (calculated at 10.60%, measured at 10.34%) are responsible for the observed mass reduction in the second step.

The final step (calculated at 70.83%, discovered at 69.56%) represents the breakdown of the remaining organic ligands. A large exothermic peak at 360 °C on the DTA curve designates this stage. The ultimate product at 450 °C is consistent with the formation of palladium oxide as a residual part was found to be (17.98%), compared to the (18.30%) calculated (Figure 6).

### 2.6. X-ray Powder Diffraction (XRD)

This XRD study is also one of the pieces of evidence about the formation of metal-ligand complexes. The manganese(II), cobalt(II), copper(II), chromium(III) and palladium(II) complexes were chosen for X-ray powder diffraction studies. The samples are irradiated with a beam of monochromatic X-rays over a variable incident angle range. The X-rays were detected using a fast-counting detector based on silicon strip technology (Bruker LynxEye detector). Interaction with atoms in the sample results in diffracted X-rays when the Bragg equation is satisfied. The X-rays were detected using a fast-counting detector based on silicon strip technology (Bruker LynxEye detector). XRD patterns are shown in Figure 7. All the samples are characterized at room temperature by X-ray diffraction using Cu Kα radiation. The diffraction pattern of complexes is recorded between 2θ ranging from 10° to 80°. The particle size of the samples is estimated using Scherrer’s formula. According to Scherrer’s equation, the particle size is given by t = 0.9 λ/Bcosθ, where t is the crystal thickness (in nm), B is half width (in radians), θ is the Bragg angle, and λ is the wavelength. The particle size corresponding to each diffraction maxima is determined from the measurement of the half-width of the diffraction peak. The value of the Lattice parameter and the particle size are shown in Table 4 for all five complexes. The Mn(II) complex is monoclinic, that of the Pd(II) and Co(II) complexes are tetragonal, and those of the Cu(II) and Cr(III) complexes belong to the triclinic crystal system. All metal complexes exhibit sharp peaks. The particle size was found to be within the range of 32–88 nm for all the complexes.

### 2.7. Morphological and Structural Properties of Synthesized Metal-Ligand Complexes

Scanning electron microscopy (SEM) was used to assess the mixed ligand metal complexes’ morphological and structural characteristics, as morphology changes when the metal ions are changed. The SEM image of Mn(II) complex revealed small particles of irregular shape. The Co(II) complex shows road-shaped particles, whereas Cr(III) complex exhibits granular texture. Finally, The Pd(II) complex shows a figure of branched nanowires. The SEM micrographs are shown in Figure 8. Thus, these SEM results confirmed the nano-structured behavior of the synthesized metal complexes.

### 2.8. Biological Activity

#### 2.8.1. In Vitro Anticancer Activities

Cancer is a type of malignant disease in which a group of cells appears to grow in an irregular pattern. The only way to treat this is through surgical intervention or chemotherapy, both of which have negative side effects. Although platinum-based complexes such as cis-platin have been used to treat cancer, it is not effective, so in vitro tests for complexes of manganese, cobalt, copper, chromium, and palladium were conducted against two cancer lines, respectively. A-549 and MRC-5 cells were used as test subjects for the cytotoxic activity of Schiff base and aminopyrimidyl complexes in the concentration range of 0–500 (g/mL). Results are provided in Figure 9, which shows the IC50 and CC50 values for each compound (Table 5). According to cytotoxicity results, all tested complexes showed strong cytotoxicity against A-549 cancer cells (IC50 = 10.97–201.86 μg/µL) and strong cytotoxicity against MRC-5 cancer cells (CC50 = 15.41–233.01 μg/µL). Copper(II) complex showed the highest cytotoxicity effect with an IC50 value of 10.97 μg/µL, followed by palladium(II) complex with an IC50 value of 26.54 μg/µL and then Cobalt(II) complex with an IC50 value of 29.94 μg/µL and then chromium(III) complex with IC50 value 61.46 μg/µL and then manganese (II) complex with IC50 value 201.86 μg/µL in case of A-549 cancer cells. Copper(II) complex showed the highest cytotoxicity effect with a CC50 value of 15.41 μg/µL, followed by palladium(II) complex with a CC50 value of 53 μM, cobalt(II) complex with a CC50 value of 101.72 μM, then chromium(III) complex with a CC50 value 157.65 μg/µL, then manganese(II) complex with CC50 value 233.01 μg/µL in case of MRC-5 cancer cells. As the concentration of the complexes increased, cell viability decreased (Figure 10).

#### 2.8.2. Antimicrobial Activity

##### Antifungal Screening

The data shown in Table 6 showed excellent activity against fungal strains, specifically against the fungal strain *Cryptococcus* nanoforms. We note that the growth inhibition zone ranged between 9 and 40. We also note that many mineral complexes showed higher activity than the control sample compared to the lower antifungal activity of the free ligand or the previous complexes [10]. We found that palladium(II) and copper(II) complexes are the most active against fungi which showed impressive activity, while the manganese(II) complex is the least effective (Figure 11, Figure 12 and Figure 13). The complexes were arranged according to their activity against fungal strains as follows: Pd(II) > Cu(II) > Co(II) > Cr(III) > Mn(II).

##### Antibacterial Screening

The data in Table 7 showed activity against bacterial strains, where the palladium(II) and copper(II) complexes showed the highest zone of inhibition compared to the rest of the complexes, and manganese(II) complexes were found to be the least effective (Figure 14). The complexes were arranged in the following order based on their activity against bacterial strains: Pd(II) > Cu(II) > Co(II) > Cr(III) > Mn(II)

Complexes are typically more effective and active against Gram-positive bacteria than Gram-negative bacteria compared to the lower antibacterial activity of the free ligand or the previous complexes [10]. This could be explained by the fact that Gram-positive bacteria have dense cell walls with many layers of teichoic acids and peptidoglycan. Gram-negative bacteria, on the other hand, have a relatively thin cell wall made up of a few peptidoglycan layers that are encircled by a second lipid membrane that contains lipoproteins and lipopolysaccharides. The variance in the structure of the cell wall led to differences in antibacterial activity. Figure 15 and Figure 16 demonstrate that the tested complexes are effective against Gram-positive and negative bacteria.

#### 2.8.3. Antioxidant Assays

Oxidation in living organisms caused by free radicals are usually undesirable processes damaging such important biomolecules as DNA, proteins, and lipids [31]. 

The application of antioxidants is one of the most straightforward means to protect these biomolecules from oxidative stress. Many assays are available for the determination of antioxidant potential in vitro. These assays are based on the reduction of stable radicals (DPPH). DPPH test is one of the most commonly performed antioxidant assays with natural and (semi)synthetic biologically active compounds. Although it has no direct physiological relevance, this assay allows quick comparison of the free radical scavenging potential of new derivatives as this activity has been published for many compounds [32]. We have therefore tested our mixed ligand complexes of Schiff bases and aminopyrimidyl derivatives for DPPH scavenging and compared their activity with the activity of the parent compounds, that is, Cu(II) and Pd(II) complexes showing high activity, Co(II) and Cr(III) complexes showing moderate activity, and all of them showing less activity when compared to the standard (ascorbic acid). Under these experimental conditions, the Mn(II) complexes demonstrated weak antioxidant activity. The IC50 values for all the collectors were determined and compared with the standard as listed in the Table 8 and Figure 17.

## 3. Materials and Methods

### 3.1. Physical Measurements for Complexes

The FTIR spectra of the ligand and its metal complexes in the solid state were measured using Agilent technologies/Gladi-ATR, USA) Cary 600 Series FTIR Spectrometer in the wave range 400–4000 cm^−1^. The carbon, hydrogen, and nitrogen contents were determined for the prepared ligand and its metallic complexes using a Eurovector CHN (EA3000, Italy) analyzer. Magnetic moments were measured using a Sherwood Scientific magnetic susceptibility balance (MSB-Auto) (UK). Electron-visible and UV-visible absorption spectra of (DMSO, 1 × 10^−3^ M) solutions were also recorded on a Shimadzu 1650-Spec UV-Vis spectrometer (Shimadzu, Duisburg, Germany). Thermal analysis of the compounds was carried out in dynamic air on a Shimadzu (DTG 60-H) thermal analyzer at a heating rate of 10 °C/min, the temperature range was 20–600 °C. XRD Model (PW 1710) control unit Philips with a Cu Kα (l = 1.54180 Å) anode at 40 K.V 30 M. A scanning electron microscopy (SEM) was used to examine the morphology of the complexes (JEOL JSM-5400-LV Field Emission SEM, Tokyo, Japan).

### 3.2. Microbial Strains and Culture Media

The complexes were tested in vitro using a diffusion agar technique; well diameter: 6.0 mm (100 μL tested) on four fungi (*Aspergillus fumigatus* (RCMB 002008), *Candida albicans* (RCMB 005003 (1) ATCC 10231), *Cryptococcus neoformas* (RCMB 0049001) and *Syncephalastrum racemosum* (RCMB 016001 (1)). The positive control was a fungicide (ketoconazole, 100 μg/mL) as well as four bacteria, two *Gram-positive* strains (*Staphylococcus aureus* (ATCC 25923) and *Bacillus subtilis* (RCMB 015 (1) NRRL B-543)), and two *Gram-negative* strains (*Escherichia coli* (ATCC 25922) and *Proteus Vulgaris* (RCMB 004 (1) ATCC 13315)). The positive control was *Streptococcus* (gentamycin, 4 μg/mL). The inhibitory activity of a compound against an organism is indicated by a clear area around the disc. 

### 3.3. Computational Studies

The density functional theory (DFT) was performed by using Gaussian09 program. The optimization of the diverse Cu(II) complex was done by DFT through the functional B3LYP and Lanl2dz/6-31G(d) basis sets. LANL2DZ basis set was limited for the treatment copper(II) atom and 6-31G(d) basis set for all other atoms. To confirm the stability of the structures, we calculated the vibrational frequencies at the same level of theory.

### 3.4. Antioxidant Activity

The antioxidant activity of the extract was determined at the Regional Center for Mycology and Biotechnology (RCMB) at Al-Azhar University by the DPPH free radical scavenging assay in triplicate, and average values were considered.

#### (DPPH) Radical Scavenging Assay 

Freshly prepared (0.004% *w*/*v*) methanol solution of 2,2-diphenyl-1-picrylhydrazyl (DPPH) radical was prepared and stored at 10 °C in the dark. A methanol solution of the test compound was prepared. About 40 uL aliquot of the methanol solution was added to 3 mL of DPPH solution. Absorbance measurements were recorded immediately with a UV-visible spectrophotometer (Milton Roy, Spectronic 1201). The decrease in absorbance at 515 nm was determined continuously, with data being recorded at 1 min intervals until the absorbance stabilized (16 min). The absorbance of the DPPH radical without antioxidant (control) and the reference compound ascorbic acid were also measured. All the determinations were performed in three replicates and averaged. The percentage inhibition (PI) of the DPPH radical was calculated according to the formula:PI = [{(AC − AT)/AC} × 100] 
where AC = absorbance of the control at t = 0 min and AT = absorbance of the sample + DPPH at t = 16 min. A 50% inhibitory concentration (IC50) or the concentration required for 50% DPPH radical scavenging activity was calculated.

### 3.5. Cytotoxicity Testing

#### 3.5.1. Mammalian Cell Lines

A-549 cells (human Lung cancer cell line) were obtained from VACSERA Tissue Culture Unit. 

#### 3.5.2. Mammalian Cell Lines

MRC-5 cells (Normal human Lung fibroblast cells), were obtained from the American Type Culture Collection (ATCC, Rockville, MD, USA).

#### 3.5.3. Chemicals Used

Dimethyl sulfoxide (DMSO), crystal violet, and trypan blue dye were purchased from Sigma (St. Louis, MO, USA). Fetal bovine serum, DMEM, RPMI-1640, HEPES buffer solution, L-glutamine, gentamycin, and 0.25% trypsin-EDTA were purchased from Lonza.

#### 3.5.4. Crystal Violet Stain (1%)

It is composed of 0.5% (*w/v*) crystal violet and 50% methanol then made up to volume with ddH_2_O and filtered through a Whatmann No.1 filter paper. 

#### 3.5.5. Cell Line Propagation

The cells were propagated in Dulbecco’s modified Eagle’s medium (DMEM) supplemented with 10% heat-inactivated fetal bovine serum, 1% l-glutamine, HEPES buffer, and 50 µg/mL gentamycin. All cells were maintained at 37 °C in a humidified atmosphere with 5% CO_2_ and were subcultured two times a week. 

### 3.6. Synthesis of the Metals Mixed Ligand Complexes

Synthesis of 2-((E)-((4-(((E)-benzylidene)amino)phenyl)imino)methyl)-nap hthalene-1-ol (HL) ligand was carried out as in the previous method [10]. Mn(II), Co(II), Cu(II), Cr(III), and Pd(II) mixed-bonding complexes were respectively prepared at a ratio of M: L1:L2 (1:1:1). 

#### 3.6.1. [Mn(HL)(ADMPY)(H_2_O)Cl_2_] (**1**)

Mn(II) complex was synthesized by dissolving MnCl_2_·4H_2_O (0.169 gm, 0.9 mmol, in 10 mL) of distilled water and adding it to an ethanolic solution of 2-((E)-((4-(((E)-benzylidene)amino)phenyl)imino)methyl)-nap hthalene-1-ol (HL) ligand (0.3 gm, 0.9 mmol, in 20 mL), followed by the addition of the second ligand 2-amino-4,6-dimethyl pyrimidine (ADMPY) (0.105 gm, 0.9 mmol, in 10 mL) of methanol. The resulting mixtures were stirred for 3 h under refluxing conditions at 60 °C. The mixture was allowed to cool to room temperature after the reflux process. A light burgundy product precipitated, which was filtered and washed with distilled water and ethanol before transferring it to the oven for drying. Elemental analysis (%) Calc. for C_30_H_29_N_5_O_2_MnCl_2_: C, 58.35; H, 4.74; N, 11.34. Found (%): C, 58.42; H, 4.09; N,11,68. Yield: (67%). Molar conductivity in DMSO, ΛM = 22.1 ohm^−1^ cm^2^ mol^−1^. M.p.: 272 °C.

#### 3.6.2. [Co(HL)(ADMPY)(H_2_O)Cl_2_]·H_2_O (**2**)

About 0.8 gm, 2.3 mmol of HL is dissolved in 30 mL of ethanol in a 250 mL beaker, then 0.54 gm, 2.3 mmol of CoCl_2_·6H_2_O is added and allowed to dissolve in 20 mL of distilled water and then continue to add. Then immediately add (0.28 gm, 2.3 mmol) of ADMPY dissolved in 10 mL of methanol. It is transferred to refluxing for 3 h. A reddish-brown precipitate was formed, and the precipitate was filtered, washed with distilled water and ethanol to remove unreacted organic materials, and then dried in the oven at 50 °C for 2 h. Elemental analysis (%) Calc. for C_30_H_31_N_5_O_3_CoCl_2_: C, 56.34; H, 4.89; N, 10.95. Found (%): C, 56.60; H, 4.97; N,10,89. Yield: (83%). Molar conductivity in DMSO, ΛM = 26.6 ohm^−1^ cm^2^ mol^−1^. M.p.: 280 °C.

#### 3.6.3. [Cu(HL)(ADMPY)(H_2_O)Cl_2_] (**3**)

It was made by combining (0.4 g, 1.1 mmol) HL Schiff base with (0.193 g, 1.1 mmol) CuCl_2_·2H_2_O, then adding ADMPY (0.140 g, 1.1 mmol) and transferred to refluxing for three hours at 60 °C. The mixture is then cooled to laboratory temperature before being filtered, washed, and dried in the oven. Finally, burgundy powder is formed. Elemental analysis (%) Calc. for C_30_H_29_N_5_O_2_CuCl_2_: C, 57.58; H, 4.68; N, 11.19. Found (%): C, 57.64; H, 4.96; N,11,85. Yield: (72%). Molar conductivity in DMSO, ΛM = 28.2 ohm^−1^ cm^2^ mol^−1^. M.p.: 266 °C.

#### 3.6.4. [Cr(HL)(ADMPY)(H_2_O)Cl_2_]·H_2_O (**4**)

About 10 mL of an aqueous solution of CrCl_3_·6H_2_O (0.304 g, 1.1 mmol) mineral salt is dissolved in a circular flask and added to 20 mL ethanolic solution of HL (0.4 gm, 1.1 mmol), then immediately add (0.140 gm, 1.1mmol) of ADMPY dissolved in 10 mL of methanol. It is transferred to refluxing for 3 h. The light brown color was produced, filtered, washed with distilled water and ethanol, and dried in the oven at 50 °C for 2 h. Elemental analysis (%) Calc. for C_30_H_31_N_5_O_3_CrCl_2_: C, 57.08; H, 4.80; N, 11.09. Found (%): C, 56.42; H, 4.98; N,11,42. Yield: (82%). Molar conductivity in DMSO, ΛM = 11.4 ohm^−1^ cm^2^ mol^−1^. M.p.: 246 °C.

#### 3.6.5. [Pd(HL)(ADMPY)(H_2_O)]·Cl_2_ (**5**)

Prepared by adding PdCl_2_ (0.3 gm, 1.3 mmol) dissolved in acetonitrile to a solution of the Schiff base ligand HL (0.468 gm, 1.3 mmol). The second ligand (0.164 gm, 1.3 mmol) was then dissolved in methanol. After 3 h of stirring and heating the mixture formed, and the dark brown precipitate was filtered and washed several times until the filters became clear, dried, and collected. Elemental analysis (%) Calc. for C_30_H_29_N_5_O_2_PdCl_2_: C, 53.88; H, 4.38; N, 10.47. Found (%): C, 53.04; H, 4.80; N,10.89. Yield: (79%). Molar conductivity in DMSO, ΛM = 12.3 ohm^−1^ cm^2^ mol^−1^. M.p.: >300 °C.

## 4. Conclusions

In summary, anticancer, antibacterial, antifungal, and antioxidant activities of previously reported of Mn(II), Co(II), Cu(II), Cr(III), and Pd(II) complexes were examined. All synthesized metal complexes showed moderate cytotoxicity against ovarian cancer cells (A-549 and MRC-5) even in the presence of DMSO in cell media. In vitro biological studies showed that the Cu(II) and Pd(II) complexes have the best anticancer activity against A-549 cells with an IC50 of 10.97 and 26.54 μM respectively, and against MRC-5 cells with a CC50 of 15.41 and 53 μM which is close to the activity of cisplatin. Out of the all complexes, the Mn(II), Co(II), and Cu(II) compounds demonstrated the highest biological activities against *G+* and *G-* bacteria in comparison to the standard drug, indicating their promising therapeutic potential. The Cu(II) and Cr(III) complexes demonstrated the highest potency against fungi. The complexes showed significant DPPH scavenging activity, and the experimental data revealed that they could be good antioxidants.

## Figures and Tables

**Figure 1 molecules-28-02555-f001:**
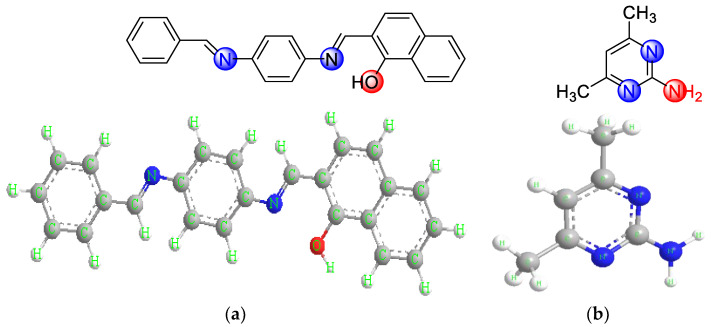
Structure of the ligands. (**a**) 2-((E)-((4-(((E) benzylidene)amino)phenyl) imino)methyl)-naphthalene-1-ol. (**b**) 2-amino-4,6-dimethyl pyrimidine.

**Figure 2 molecules-28-02555-f002:**
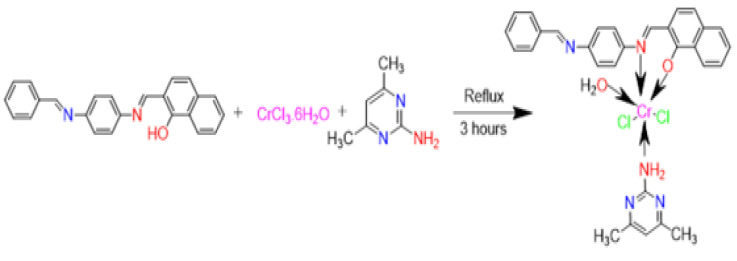
Preparation of the Cr(III) complex (**4**).

**Figure 3 molecules-28-02555-f003:**
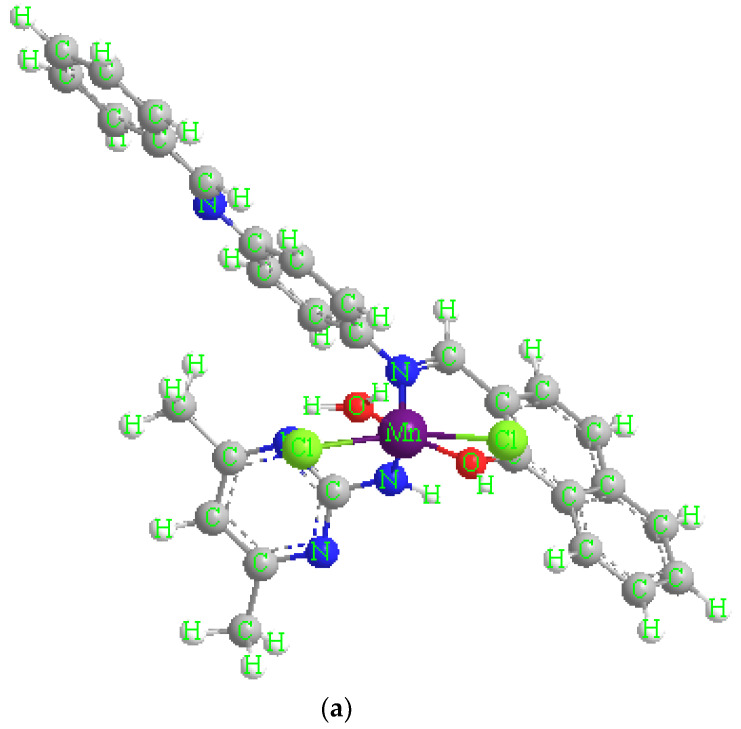
A perspective view of the complete coordination around (**a**) Mn(II) complex. (**b**) Pd(II) complex.

**Figure 4 molecules-28-02555-f004:**
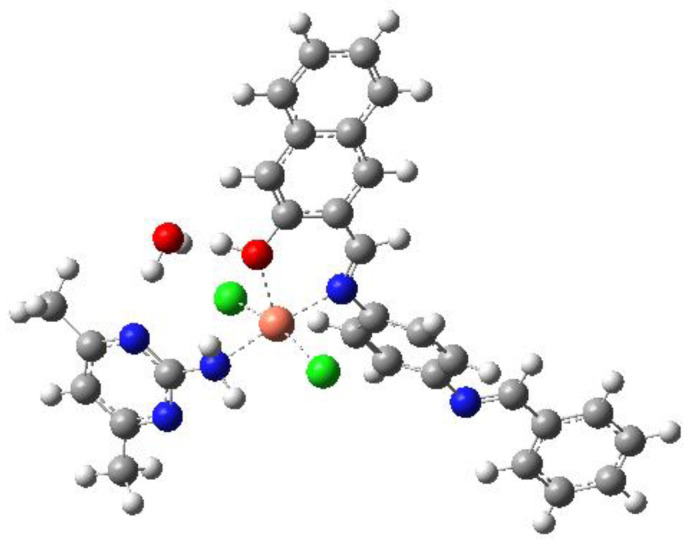
DFT Optimized structure of Cu(II) complex model.

**Figure 5 molecules-28-02555-f005:**
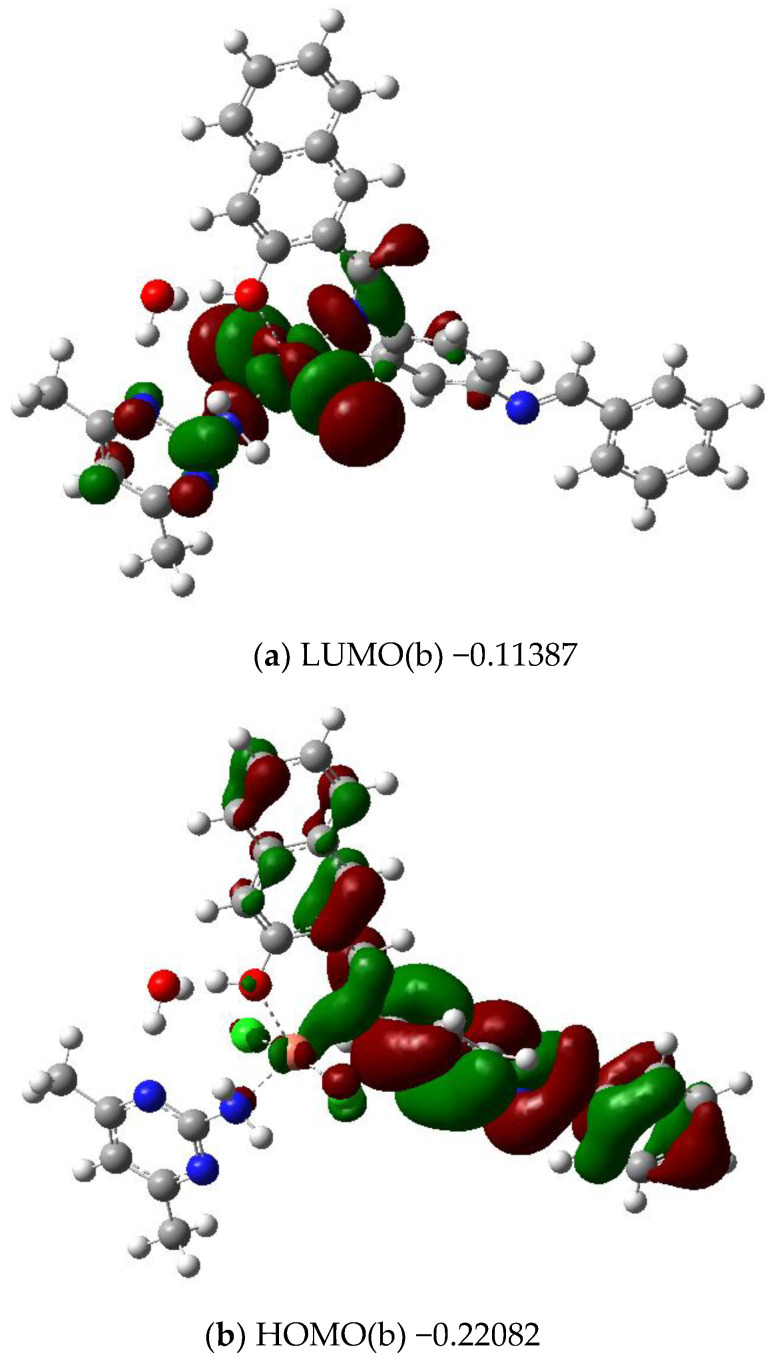
Frontier orbitals and their energies for the Cu(II) complex (**a**,**b**).

**Figure 6 molecules-28-02555-f006:**
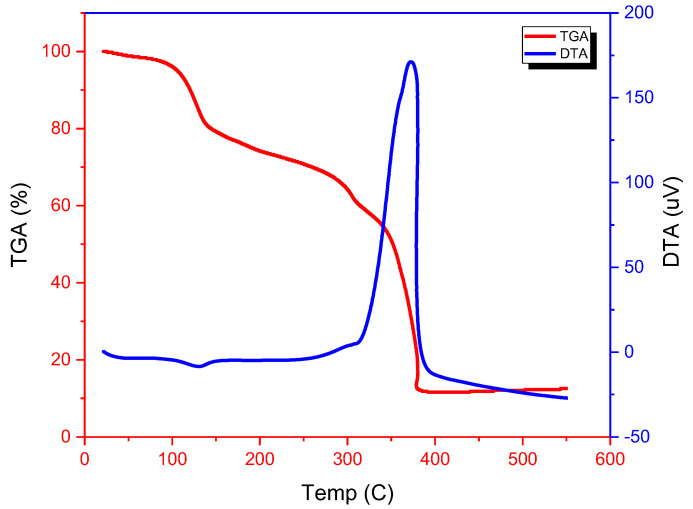
TGA and DTA thermograms of Pd(II) complex.

**Figure 7 molecules-28-02555-f007:**
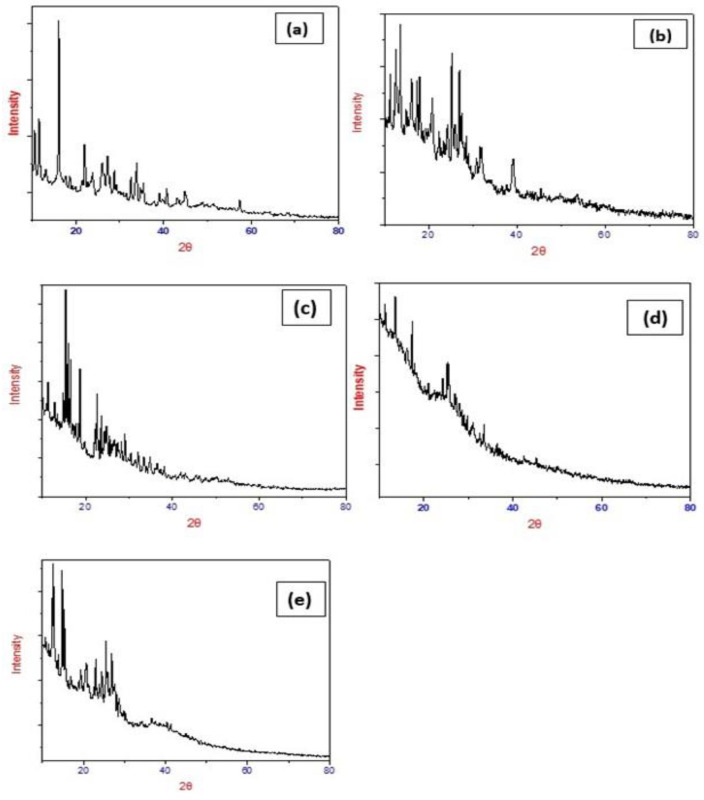
X-ray powder diffraction pattern for the (**a**) Mn(II), (**b**) Co(II), (**c**) Cu(II), (**d**) Cr(III), and (**e**) Pd(II) complexes.

**Figure 8 molecules-28-02555-f008:**
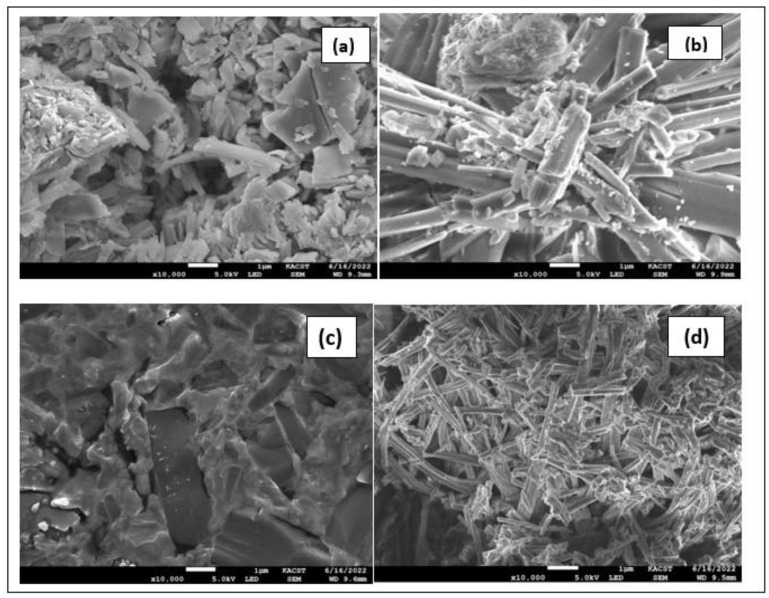
SEM micrographs of (**a**) Mn(II), (**b**) Co(II), (**c**) Cr(III), and (**d**) Pd(II) complexes.

**Figure 9 molecules-28-02555-f009:**
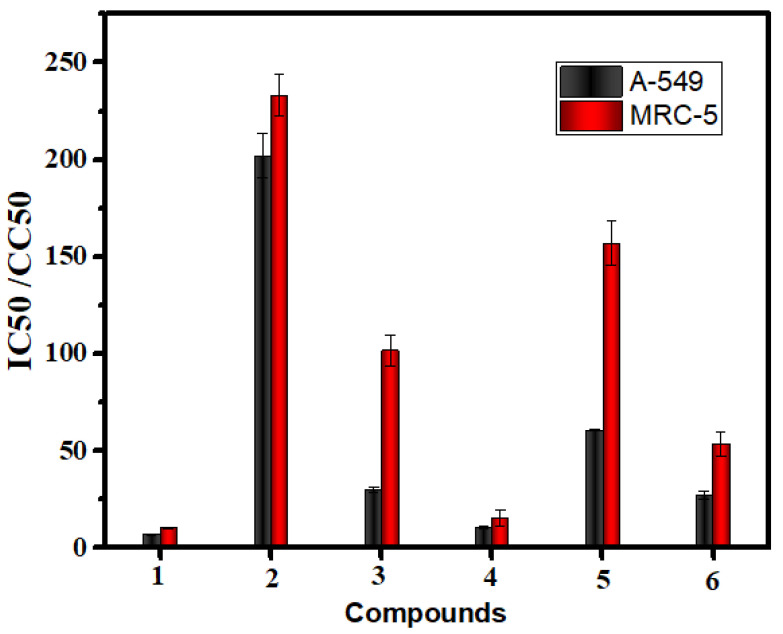
The comparison of anticancer activity for complexes (**2**–**6**) with standard drug cisplatin against (**1**) against selected human cell lines.

**Figure 10 molecules-28-02555-f010:**
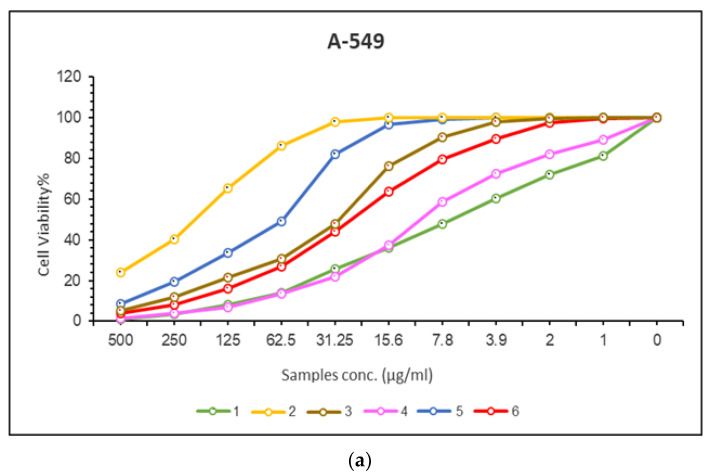
(**a**) A-549. The dose-response curves of the cytotoxicity of compounds (**1**–**6**). Cells were exposed to different concentrations of the compounds for 72 h. Cell viability was determined by a colorimetric method. (**b**) MRC-5cell lines. The dose-response curves of the cytotoxicity of compounds (**1**–**6**) Cells were exposed to different concentrations of the compounds for 72 h. Cell viability was determined by a colorimetric method.

**Figure 11 molecules-28-02555-f011:**
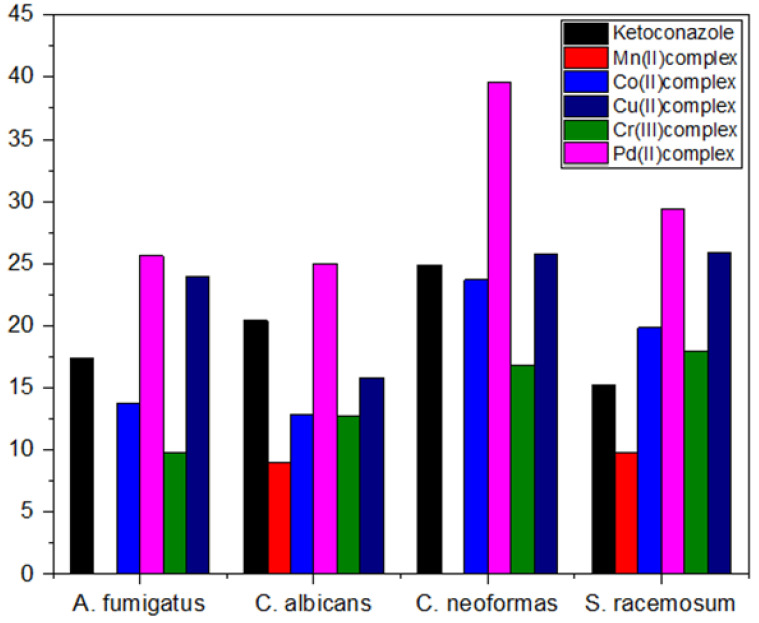
Antifungal activities of the Ketoconazole and five metal complexes.

**Figure 12 molecules-28-02555-f012:**
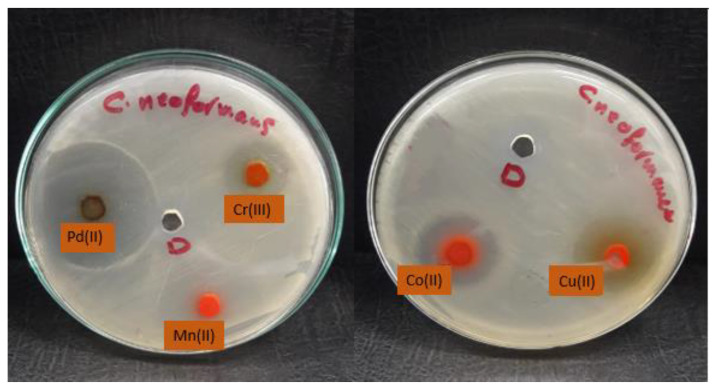
Microbiological screening of coordination compounds of Mn(II), Co(II), Cu(II), Cr(III), and Pd(II) toward *Cryptococcus neoformas*.

**Figure 13 molecules-28-02555-f013:**
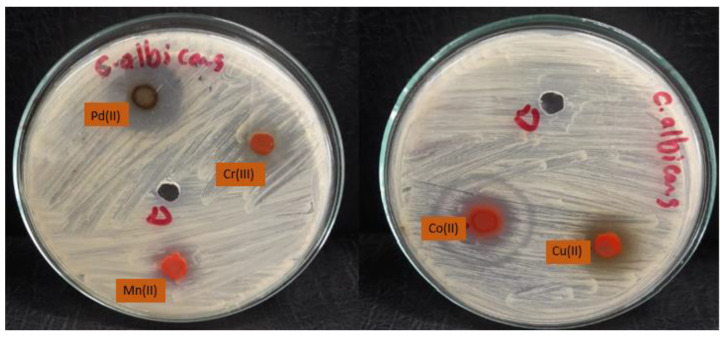
Microbiological screening of coordination compounds of Mn(II), Co(II), Cu(II), Cr(III), and Pd(II) toward *Candida albicans*.

**Figure 14 molecules-28-02555-f014:**
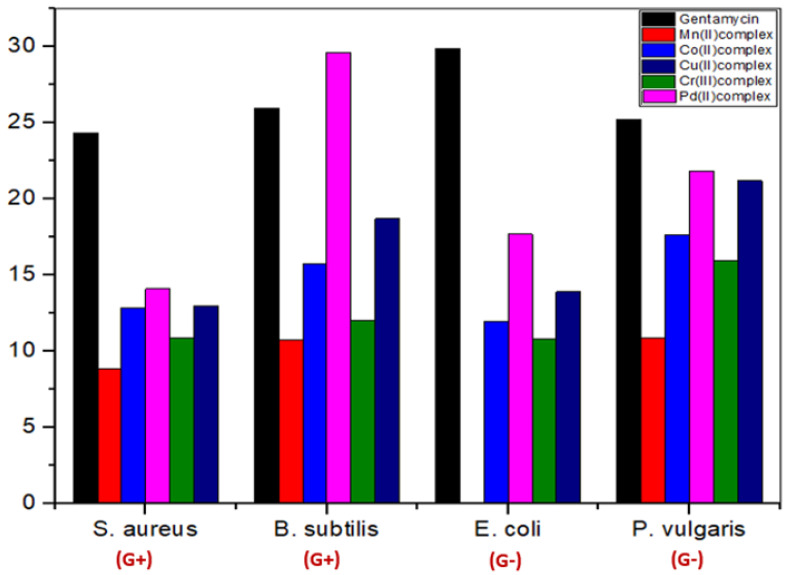
Antibacterial activities of the gentamycin and five metal complexes against *S. aureus, B. subtilis, Escherichia coli* and *P. vulgaris*.

**Figure 15 molecules-28-02555-f015:**
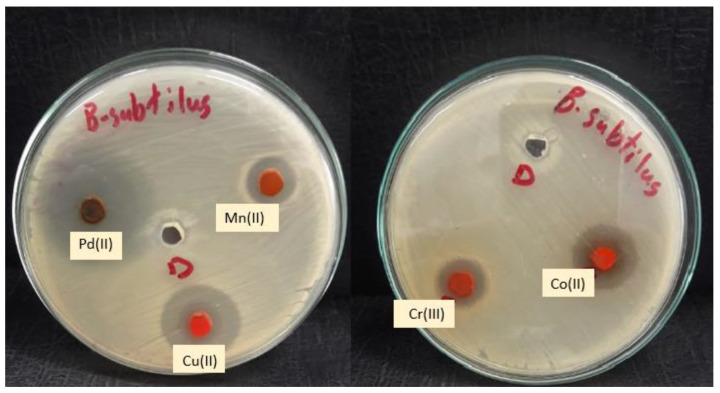
Microbiological screening of coordination compounds of Mn(II), Co(II), Cu(II), Cr(III), and Pd(II) toward *Bacillus Subtilis*.

**Figure 16 molecules-28-02555-f016:**
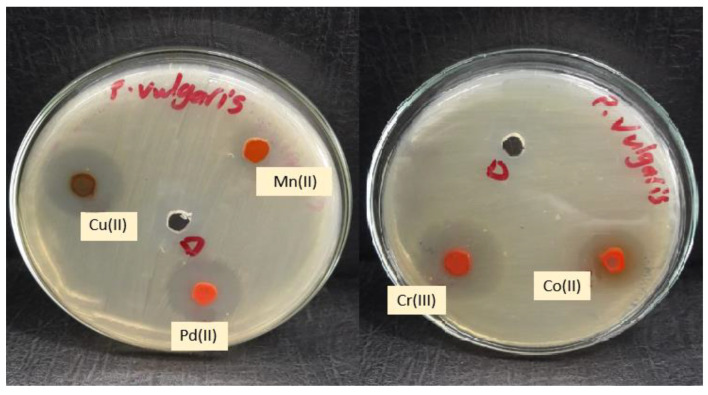
Microbiological screening of coordination compounds of Mn(II), Co(II), Cu(II), Cr(III), and Pd(II) toward *Proteus vulgaris*.

**Figure 17 molecules-28-02555-f017:**
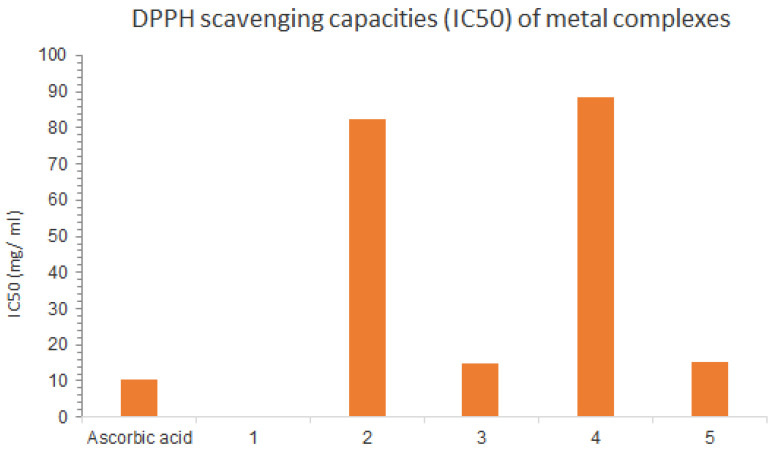
Variation of IC50 values for all complexes and standards.

**Table 1 molecules-28-02555-t001:** FT-IR details spectral data (cm^−1^) of free ligands and its metal compounds.

Group	(HL)	(ADMPY)	1	2	3	4	5
υ(OH)	3350	-	3410	3345	3438	3326	3354
υ(OH) lattice water	-	-	-	3547	-	3548	-
υ(OH) coordinated water	-	-	3330	3305	3314	3312	3302
υ_as/s_ (NH_2_)	-	3396, 3310	Disappear	Disappear	Disappear	Disappear	Disappear
δ(NH_2_)	-	1633	1680	1666	1680	1668	1698
υ(C=N)	1618, 1604	-	1616, 1548	1625, 1592	1617, 1550	1613, 1588	1610, 1575
δ(OH) in-plane	850	-	834	826	833	821	819
υ(C=C)	1540	-	1530	1509	1526	1538	1536
υ(C-O)	1310	-	1297	1316	1297	1309	1305
υ(M-O)	-	-	640	612	634	626	620
υ(M-N)	-	-	566	547	560	554	544
υ(M-Cl)	-	-	442	419	440	430	424

υ: stretching, δ:bending, υ_as_: antisymmetric stretching υ_S_: symmetric stretching; (HL): C_24_H_18_N_2_O, (ADMPY): C_6_H_9_N_3_; **1**: C_30_H_29_N_5_O_2_MnCl_2_; **2**: C_30_H_31_N_5_O_3_CoCl_2_; **3**: C_30_H_29_N_5_O_2_CuCl_2_; **4**: C_30_H_31_N_5_O_3_CrCl_2_; **5**: C_30_H_29_N_5_O_2_PdCl_2_

**Table 2 molecules-28-02555-t002:** Electronic spectral data of free ligand and its complexes and magnetic moments of the compounds.

Ligands and Complexes	_Vmax_ (nm)	v_max_ (cm^−1^)	Assignment	*μ*eff. (B.M)	Geometry	d^n^
(HL)	259	38,610	*π*→*π*∗	-	-	-
(ADMPY)	314	31,847	*n*→*π*∗	-	-	-
Mn(II) complex	270	37,037	*π*→*π*∗	5.84		
	319	31,347	*n*→*π*∗		Octahedral	d^5^
	455	21,978	d-d transition			
Co (II) complex	252	39,682	*π*→*π*∗	4.90		
	309	32,620	*n*→*π*∗		Octahedral	d^7^
	426	23,474	d-d transition			
Cu(II) complex	261	38,314	*π*→*π*∗	1.72		
	300	33,333	*n*→*π*∗		Octahedral	d^9^
	415	24,096	d-d transition			
Cr(III) complex	295	33,898	*π*→*π*∗	3.70		
	345	28,985	*n*→*π*∗		Octahedral	d^3^
	419	23,866	d-d transition			
Pd(II) complex	308	32,468	*π*→*π*∗	Diamagnetic		
	355	28,169	*n*→*π*∗		square planar	d^8^
	454	22,026	d-d transition			

*n* = number of electrons.

**Table 3 molecules-28-02555-t003:** Thermal decomposition data of complexes.

Compound	Step	Temp. Range °C	Assignment	TGA (Wt. loss %) Found (calcd.)
**Mn(II) complex**	1st 2nd 3rd 4th	80–140 142–200 200–302 304–550	Loss of coordinated water molecules (H_2_O) Loss of two chloride atoms Loss of ligand (ADMPY) Decomposition the rest of the organic ligand (HL) final product (manganese oxide)	2.59 (2.91) 11.30 (11.48) 19.72 (19.94) 55.86 (56.75) 10.53 (11.48)
**Co(II) complex**	1st 2nd 3rd 4th	78–150 152–248 250–396 398–550	Loss two water molecules, one crystalline and one coordinated. Loss of two chloride atoms Loss of ligand (ADMPY) Decomposition with the formation of final product (cobalt oxide).	5.40 (5.62) 10.98 (11.08) 19.10 (19.26) 53.11 (54.80) 15.70 (15.93)
**Cu(II) complex**	1st 2nd 3rd 4th 5th	83–154 156–270 272–362 364–420 422–550	Loss of coordinated water molecules (H_2_O) Loss of two chloride atoms Loss of ligand (ADMPY) Loss of ligand (HL) formation of copper oxide.	2.67 (2.87) 10.96 (11.33) 19.47 (19.68) 55.80 (56.01) 12.02 (12.71)
**Cr(II) complex**	1st 2nd 3rd 4th	81–158 160–221 223–353 355–550	Loss two water molecules, one crystalline and one coordinated. Loss of two chloride atoms Loss of ligand (ADMPY) Decomposition with the formation of final product chromium oxide.	5.13 (5.68) 10.98 (11.21) 19.14 (19.48) 54.87 (55.38) 10.56 (10.75)
**Pd(II) complex**	1st 2nd 3rd	82–160 162–254 256–550	Loss of coordinated water molecules (H_2_O) Loss of two chloride atoms Thermal decomposition of the rest of the complex and forming palladium oxide.	2.51 (2.69) 10.34 (10.60) 69.56 (70.83) 17.98 (18.30)

**Table 4 molecules-28-02555-t004:** X-ray powder diffraction crystal data of complexes.

Parameters	Mn(II) Complex	Co(II) Complex	Cu(II) Complex	Cr(III) Complex	Pd(II) Complex
Empirical formula	C_30_H_29_N_5_O_2_MnCl_2_	C_30_H_31_N_5_O_3_CoCl_2_	C_30_H_29_N_5_O_2_CuCl_2_	C_30_H_31_N_5_O_3_CrCl_2_	C_30_H_29_N_5_O_2_PdCl_2_
Formula weight	617.42	639.42	625.72	632.16	668.60
Crystal system	Monoclinic	Tetragonal	Triclinic	Triclinic	Tetragonal
a (Å)	17.160	12.550	7.044	5.382	13.281
b (Å)	5.861	12.550	10.298	3.345	13.281
c (Å)	16.840	13.090	13.419	14.532	6.228
Alfa (°)	90.000	90.000	55.914	91.749	90.000
Beta (°)	101.310	90.000	71.226	26.757	90.000
gamma (°)	90.000	90.000	50.082	89.817	90.000
Volume of unit cell (Å3) Particle Size (nm)	1661 72	2064 67	617.34 38	117.58 32	1098.6 88

**Table 5 molecules-28-02555-t005:** The inhibitory effects (IC50%) (CC50%) of cisplatin compound and Mn(II), Co(II), Cu(II), Cr(III), and Pd(II) complexes against different types of cell lines.

No.	Compounds	^a^ A-549		^b^ MRC-5	
		^c^ IC50 Value (µg/mL)	±S.D	^d^ CC50 Value (µg/mL)	±S.D
1	cisplatin	7.11	0.39	10.51	0.37
2	Mn(II) complex	201.86	11.48	233.01	10.74
3	Co(II) complex	29.94	1.14	101.72	8.26
4	Cu(II) complex	10.97	0.67	15.41	4
5	Cr(III) complex	61.46	0.21	157.65	11.28
6	Pd(II) complex	26.54	2.08	53	6.09

^a^ Human lung cancer cell line, ^b^ normal human lung fibroblast cells, ^c^ average IC50 values from three independent experiments for drug concentration μg/mL of 50% cell death following 72 h exposure, ^d^ the concentration required to cause toxic effects in 50% of intact cells.

**Table 6 molecules-28-02555-t006:** Biological activities of the complexes against fungal.

Mean Zone of Inhibition in mm for Three Replicates								
Compounds	*Aspergillus fumigatus*		*Candida albicans*		*Cryptococcus neoformas*		*Syncephalastrum racemosum*	
^1^ Ketoconazole	17.37	±0.55	20.40	±0.82	24.87	±0.31	15.27	±0.57
Mn(II) complex	^2^ NA	-	8.97	± 0.21	NA^2^	-	9.80	±0.36
Co(II) complex	13.73	±0.32	12.90	±0.26	23.67	±0.47	19.87	±0.38
Cu(II) complex	23.97	±0.47	15.80	±0.36	25.73	±0.40	25.87	±0.21
Cr(III) complex	9.83	±0.32	12.77	±0.40	16.83	±0.31	17.90	±0.30
Pd(II) complex	25.63	±0.49	25	±0.46	39.57	±0.57	29.40	±0.44

^1^ control of antifungus, ^2^ no activity.

**Table 7 molecules-28-02555-t007:** Biological activities of the complexes against Gram-positive bacteria.

Mean Zone of Inhibition in mm for Three Replicates								
(G+) Gram-Positive Bacteria (G−) Gram-Negative Bacteria								
Compounds	*Staphylococcus aureus*		*Bacillus subtilis*		*Escherichia coli*		*Proteus vulgaris*	
^3^ Gentamycin	24.33	±0.15	25.93	±0.32	29.83	±0.32	25.20	±0.26
Mn(II) complex	8.87	±0.42	10.73	±0.42	^4^ NA	-	10.85	±0.35
Co(II) complex	12.83	±0.42	15.77	±0.42	11.93	±0.35	17.60	±0.62
Cu(II) complex	13	±0.46	18.70	±0.44	13.90	±0.26	21.20	±0.40
Cr(III) complex	10.87	±0.31	12	±0.40	10.80	±0.52	15.93	±0.55
Pd(II) complex	14.10	±0.26	29.60	±0.52	17.67	±0.60	21.83	±0.42

^3^ Control of antibacterial, ^4^ no activity.

**Table 8 molecules-28-02555-t008:** IC50 values for the antioxidant activities of the mineral complexes and the standard.

Compounds	IC_50_ (mg/mL)
Ascorbic acid	10.21 ± 0.77
Mn(II) complex	weak antioxidant activity
Co(II) complex	82.43 ± 1.85
Cu(II) complex	13.98 ± 0.52
Cr(III) complex	87.76 ± 5.11
Pd(II) complex	15.03 ± 0.36

## Data Availability

By e-mail: m.alfakeh@qu.edu.sa.

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
