# Peer review of "Physico-Chemical Study of Mn(II), Co(II), Cu(II), Cr(III), and Pd(II) Complexes with Schiff-Base and Aminopyrimidyl Derivatives and Anti-Cancer, Antioxidant, Antimicrobial Applications"

_molecules, 2023, doi:10.3390/molecules28062555_

Round 1

Reviewer 1 Report

Manuscript Molecules-2263060

Authors: Maged S. Al-Fakeh, Maha A. Alsikhan and Jawza Sh Alnawmasi

Title: Physico-chemical Study of new nano-sized Mn(II), Co(II), Cu(II), Cr(III) and Pd(II) complexes with Schiff-Base and Ami-nopyrimidyl derivatives and Anti-cancer, Antioxidant, Antimi-crobial Applications

This paper describes the synthesis and characterisation of new complexes with mixed ligands.

The major observation is the lack of any crystal structure which, in my opinion, is mandatory for such journal.

The Schiff base used for the synthesis of the complexes was synthesized according to ref. 10. In ref. 10 shows a scheme in which a symmetrical diamine reacts with two different carbonyl compounds, each with an amino group. There is no reason for such a reaction. Usually, both amino groups react with the same carbonyl compound, resulting in a mixture of Schiff bases. There is no evidence for the purity of the Schiff base.

Still, assuming that the authors obtained the Schiff base, the formulation of complexes is unrealistic:

-        The OH group from Schiff base usually deprotonates in complexes, as IR spectra confirm for reported complexes;

-        The Cr(III) complex is not neutral, according to proposed formula (only two chloride anions);

-        There is no IR and UV-Vis spectra presented;

-        the Electronic spectra chapter contains only spectra recorded in DMSO solution. It is well known that different interactions with solvent molecules can occur in solution, therefore the stereochemistry of the complexes could be changed. In the absence of single-crystal X-ray diffraction, the only information regarding stereochemistry will be provided by solid-state electronic spectra.

-        In manganese complex derivatogram the total mass loss is about 45%. In Table 1 the total mass loss is 89,47% (as it should be!!!).

Based on these comments, I propose to reject presented work. 

Author Response

Dear Professor

Thank you so much for your sending great notes.

Best regards

Reviewer 2 Report

Authors presented  results of studies of several metal complexes based on mixed ligands: Schiff base, derivative of 2-hydroxy-1-naphthaldehyde and aminopirymidyl. I have some questions and remarks which have to be considered before publication in Molecules.

1.  First at all I have serious doubt concerning structure of new compounds-it is fundamental question concerning object of the studies. Obviously, Authors obtained metal complexes, but which ones? Based on what data Authors proposed structure of the obtained complexes where –OH group of Schiff base remains in complex as –OH not –O-Me?  EA data? FT-IR data? Typically new O-Me bond is formed for such type complexes, not OH-Me bond. In my opinion presence of the d(OH) rocking (by the way-position of this band  should be defined in text or in Table 1) is not the proof of it, especially that assignment of OH bands due to presence of several water molecules is very problematic. For Pd(II) complex Authors can easily prove it comparing H NMR spectra of the HL ligand and complex. Presence of OH signal at ca. 12-14 ppm in the NMR spectrum of complex will indicate suggested –OH-Me structure of the Pd(II) complex (however, it can require dry DMSO). By the way, Figure 4b (Figure 4a also) shows complex without hydrogen at –OH group.  Another question arise for Cr(III) complex-proposed structure is [Cr(HL)(ADMPY)(H2O)Cl2]H2O. But chromium is Cr(III)-so two Cl- ions is not enough.

2. Authors have to show at least one example of recorded IR and UV-Vis spectra, I mean comparison of the ligand and complex. Good idea is to  attach supplementary materials with all spectra, thermograms, etc.

3. In my opinion data for HL and ADMPY presented in Table 2 are incomplete. UV-Vis spectrum of HL included in reference 10 shows more bands that presented-band at ca. 375 nm is missing. Presence of this band can indicate the existence of proton transfer equilibrium.

4. In Introduction Authors should describe more precisely importance of the 2-amino-4,6-dimethylpyrimidine (or its derivatives) in biological systems and more recent references should be mentioned (e.g. in J.Mol.Struct. 2023).

5. I have also question concerning X-ray data in Table 4 about accuracy of the calculated parameters of  unit cell which are not presented.

6. Authors in Introduction mentioned that they want compare biological activity of complexes based on HL (Reference 10) with those obtained within this work based on HL and ADMPY. However in the manuscript I can not find any comparison. In my opinion Authors have to include results of biological activity for previously studied (e.g. from mentioned reference 10) compounds and for pure ligands HL and ADMPY. Right now for  readers are available data for complexes 1-5 only. This part of the manuscript have to be deepen and complete, e.g. comparison of biological activities of complexes with and without ADMPY (due to the one of the main aims of this work). I also strongly suggest to include data of biological activity of pure HL ligand and ADMPY.   

7. In Conclusions  phrases concerning nanotechnology have to be removed. Synthesis of the studied nano-size complexes in my opinion is not nanotechnology at all and presented results does not support such conclusions and are misuse! Authors obtained nano-size complexes accidentally, as a result of typical synthesis. Only if Authors can obtain complexes larger than nano-size or/and using more sophisticated method, they compare results for both series of compounds and  observe significant differences-it can allow them to write such sentences.    

Author Response

Dear Professor

Thank you so much for sending great notes.

Best regards 

Round 2

Reviewer 1 Report

After revision the paper was improved.

However, the thermal analysis remains unresolved, especially for the manganese complex. From the DTA curve (the red one) the decomposition starts at 100% and ends at approximately 55%. So the total loss is 100-55=45%!!!!!!!!!!

It is clear to me that thermal decomposition, except for the Pd complex, ends at temperatures greater than 550°C.

Author Response

Dear Professor

I hope you are fine.

Best regards 

Reviewer 2 Report

Authors in revised version of the manuscript have taken into account some of my remarks, but some (most important) not.

1.  Again, I still have doubts concerning object of this studies. In my opinion, they are still undefined. I agree, Authors obtained new complexes, but proposed structure of them can be suggestions only. I can not agree for structures with –OH-Me bonds. Only if Authors have results of HR MS (with four digits accuracy) or resolved crystal structure by X-ray I can agree with Authors.

 2. Carefully inspection of the EA analysis shows that for Mn(II) complex the difference between data calculated and found for H is really high (ca. 0.65%). When one hydrogen is not taken into account, I mean –O-Me bond is forming this difference is 0.15% only. In this case, values for other elements are quite comparable.  For other complexes similar situation occur. I agree, it makes trouble when Authors try to fit EA data to possible structure of the complexes. But Authors did not perform EA measurements for amount of halogens and presence of halogens can strongly influence results for other elements.

3. Figure 4a and Figure 4b show complexes without hydrogen at –OH group, so how Authors can still suggest presence of the –OH-Me bond? 

3. IR spectrum of ligands (it should be mentioned that these spectra are ATR IR spectra, I guess) shows weak band at ca. 3000 cm-1 which can be assigned to the intramolecular hydrogen bond. This band disappeared in all complexes. Presence of water molecules make this picture even more complicated. By the way, Authors  write (page 7, top) “This chelation is further reinforced also by appearance of new peaks in complexes due to δ(OH) rocking[18], its blue-shift indicates the sharing of phenolic oxygen atom in coordination with metal ions after losing its proton.

4. In Table 2 UV-Vis bands for HL are still incomplete. Where are at least two bands at ca. 400, 474 nm which are visible in UV-Vis spectrum?

5. In Table 4 still accuracy of the calculated parameters  are not included. Why for some parameters are 3 digits while for some 2 digits only?

6. Because I understand problems with establishing real structure of obtained complexes and  because no additional measurements can be made, I suppose, (e.g. HR MS or others) I have suggestion which can help. First, Authors have to modify  Figure 2.  Authors have to keep structure for Cr(III) complex which without doubts has –O-Me bond (Figure 2b) as example of the synthesis only (other examples have to be removed) and remove Fig. 3. In manuscript (page 4 and 5) at the end of this paragraph phrase that these are suggested summary formulas only must be added.  

7.  Another problem is with DFT calculations. However, such calculations were carried out for one complex only. In my opinion,  they did not contribute anything significant, so they can be omitted without loss of the significance for the manuscript.

Author Response

(The authors gave the same response as above.)
